# ReHub: Linear Complexity Graph Transformers with Adaptive Hub-Spoke Reassignment

## Abstract

We present ReHub, a novel graph transformer architecture that achieves linear complexity through an efficient reassignment technique between nodes and virtual nodes. Graph transformers have become increasingly important in graph learning for their ability to utilize long-range node communication explicitly, addressing limitations such as oversmoothing and oversquashing found in message-passing graph networks. However, their dense attention mechanism scales quadratically with the number of nodes, limiting their applicability to large-scale graphs. ReHub draws inspiration from the airline industry's hub-and-spoke model, where flights are assigned to optimize operational efficiency. In our approach, graph nodes (spokes) are dynamically reassigned to a fixed number of virtual nodes (hubs) at each model layer. Recent work, Neural Atoms (Li et al., 2024), has demonstrated impressive and consistent improvements over GNN baselines by utilizing such virtual nodes; their findings suggest that the number of hubs strongly influences performance. However, increasing the number of hubs typically raises complexity, requiring a trade-off to maintain linear complexity. Our key insight is that each node only needs to interact with a small subset of hubs to achieve linear complexity, even when the total number of hubs is large. To leverage all hubs without incurring additional computational costs, we propose a simple yet effective adaptive reassignment technique based on hub-hub similarity scores, eliminating the need for expensive node-hub computations. Our experiments on long-range graph benchmarks indicate a consistent improvement in results over the base method, Neural Atoms, while maintaining a linear complexity instead of $O(n^{3/2})$. Remarkably, our sparse model achieves performance on par with its non-sparse counterpart. Furthermore, ReHub outperforms competitive baselines and consistently ranks among the top performers across various benchmarks.

## 1 Introduction

Learning on graphs is essential in numerous domains, including social networks for influence prediction, biological networks for understanding protein interactions, molecular graphs for predicting chemical properties, knowledge graphs for recommendation systems, and financial networks for fraud detection and risk assessment. Graph neural networks (GNNs) have emerged as powerful tools in these areas, operating via message passing between connected nodes. However, a significant challenge with GNNs is their limited communication range. While stacking message-passing layers can increase the communication distance, it comes at a computational cost and can cause issues such as oversmoothing and oversquashing (Alon & Yahav, 2020; Topping et al., 2021).

Inspired by the success of transformers in natural language processing (Vaswani et al., 2017), graph transformers offer a solution by enabling global node communication through attention mechanisms (Dwivedi & Bresson, 2020; Shehzad et al., 2024). This overcomes the communication bottlenecks of GNNs, but it comes at a significant computational cost. The quadratic complexity of dense attention operations limits the scalability of graph transformers, as even modest-sized graphs can exhaust GPU memory. Several methods have been suggested to reduce the complexity of global attention. For example, GraphGPS (Rampášek et al., 2022) combines sparse attention mechanisms like Performer (Choromanski et al., 2020) or Big Bird (Zaheer et al., 2020). Originally designed for processing sequences rather than graph structure, these linear-memory transformers induce significant computational time and do not match the performance of dense attention (Shirzad et al., 2023).

Recently, transformer-based graph networks have utilized the addition of virtual global nodes, through which graph nodes communicate to sparsify the attention. By constraining the attention to be between the graph nodes and these virtual nodes, the attention complexity is reduced to the number of nodes times the number of virtual nodes, allowing the overall complexity to be governed by the number of virtual nodes. Exphormer (Shirzad et al., 2023) maintains linear complexity by using a fixed number of virtual nodes, while Neural Atoms (Li et al., 2024) explores both a fixed number and a ratio relative to the number of nodes. An important finding in Neural Atoms is that adding more virtual nodes increases prediction accuracy, creating a trade-off between computational complexity and accuracy.

In this work, we introduce ReHub, a novel graph transformer architecture that achieves linear complexity by dynamically reassigning graph nodes to virtual nodes. We are inspired by complex systems where efficient connectivity and adaptability are crucial for optimal performance. A pertinent example is the airline industry, where flights are dynamically assigned to a limited number of major airports (hubs) to optimize operational efficiency. We identify the graph nodes with spokes and the virtual nodes with hubs. The key insight of our approach stems from noting that the transformer's complexity is driven by spoke-hub attention. Therefore, it is not necessary to reduce the total number of hubs to achieve linear complexity; instead, using a fixed, small number of connected hubs per spoke is sufficient. To effectively utilize all hubs without increasing computational cost, we introduce a simple yet efficient adaptive reassignment mechanism. In order to so and yet avoid the costly computation the entire set of spoke-hub interactions, our reassignment mechanism is based on hub-hub similarity scores, which are cheap to compute.

In summary, then, our primary contribution is a novel graph transformer architecture that integrates global attention with an efficient spoke-hub reassignment strategy, significantly enhancing scalability while maintaining performance. Our experiments on long-range graph benchmarks indicate a consistent improvement in results over the base method, Neural Atoms, while keeping a linear complexity instead of $O(n^{3/2})$. Remarkably, our sparse model achieves performance on par with its non-sparse counterpart. Furthermore, ReHub outperforms competitive baselines and consistently ranks among the top performers across various benchmarks.

## 2 RELATED WORK

**Learning on large graphs** Graph learning architectures are a well-established and highly active field of research (Wu et al., 2020). Common GNNs, such as GCN/GCN2 (Kipf & Welling, 2016; Chen et al., 2020), GAT/GATv2 (Velickovic et al., 2017; Brody et al., 2021), GIN/GINE (Xu et al., 2018; Hu et al., 2019), and GatedGCN (Bresson & Laurent, 2017), rely on a message-passing architecture that aggregates information. In each layer every graph node updates its representation by aggregating the neighboring nodes. This architecture inherently limits their ability to accumulate information over large distances due to phenomena such as over-smoothing (Alon & Yahav, 2020), where node representations become indistinguishable, and over-squashing (Topping et al., 2021) , where the capacity to propagate information is restricted by bottlenecks. Consequently, learning on large graphs remains a persistent challenge (Duan et al., 2022). To address this issue, some methods focus on reducing the memory footprint. One approach involves dividing graphs into mini-batches (Wu et al., 2024), while another uses only segments of the graph for training (Cao et al., 2024).

Transformer architectures have recently gained popularity for graph-based tasks (Müller et al., 2023). These methods address the over-smoothing and over-squashing issues by enabling all nodes to interact with each other through attention (Velickovic et al., 2017; Ying et al., 2021). However, this approach is computationally inefficient, with quadratic time and memory consumption in the number of nodes. To mitigate these inefficiencies, more efficient transformer architectures have emerged which utilize different approaches to reduce the number of computations such as approximations in Performer (Choromanski et al., 2020), predefined attention patterns in BigBird (Zaheer et al., 2020) and better parallelism and partitioning in FlashAttention (Dao et al., 2022). GraphGPS (Rampášek et al., 2022) proposes a general framework for combining message-passing neural networks (MPNNs) with attention at each layer, facilitating the use of such attention mechanisms.

Recent works have introduced approaches that leverage the structure and inherent information of the graph, such as tokenization of hops of node's neighbors in NAGphormer and applying structure-preserving attention on encoded sequences of sub-graphs in Gophormer

(Chen et al., 2022; Zhao et al., 2021). Some of these works ensure that the computational complexity remains linear in the number of nodes (Shirzad et al., 2023; Chen et al., 2022; Wu et al., 2024; 2022) which allows them to be applied to larger graphs. We also propose a linear complexity architecture that passes nodes' information through a medium that efficiently orchestrates the passage of information in the graph and between nodes.

Finally, we note that recently, State Space Models (SSMs) (Gu & Dao, 2023) have emerged as a promising approach for efficiently processing large sequences, with their adaptation to graphs showing notable results (Wang et al., 2024; Behrouz & Hashemi, 2024). However, in this work, we focus on transformer-based architectures, which remain the current go-to approach.

**Virtual nodes** The concept of virtual nodes involves introducing new external nodes that interact with the graph to facilitate information exchange between existing nodes (Gilmer et al., 2017). Recent studies (Shirzad et al., 2023; Cai et al., 2023; Hwang et al., 2022) utilize this concept to extend the graph's capability to capture long-range information through message-passing. Other research explores the integration of virtual nodes within the context of transformers (Shirzad et al., 2023; Li et al., 2024; Fu et al., 2024). Similarly, we use virtual nodes that communicate with both each other and the graph nodes through attention. However, unlike prior methods, we dynamically update the connectivity between the virtual nodes and graph nodes at each layer, enhancing the flow of information.

## 3 METHOD

### 3.1 PIPELINE OVERVIEW

Our proposed architecture, depicted in Figure 1, is aimed at handling long-range graph node communication while maintaining computational efficiency. This is implemented through hubs, that act as information aggregators and distributors from and to the spokes. This allows for long range communication to be manifested as hub-hub communication. ReHub is carefully designed to follow our key observation that the complexity can be kept linear as long as: (1) the number of hubs $N_h$ is kept small enough, on the order of $\sqrt{N_s}$, where $N_s$ is the number of spokes; and (2) $k$, the number of hubs connected to each spoke per layer, is a small constant (*e.g.*, $k = 3$).

In what follows we describe the architecture of ReHub, define every component and the interaction between spokes and hubs. First we present an overview of the notation. Then we present an initialization scheme for the hubs and explain each part of the architecture: (1) Spokes-Spokes update (2) Spokes-Hubs update (3) Hubs-Hubs update (4) Hubs-Spokes update (5) Hub (Re)Assignment. Finally, we show that the complexity given by this architecture is linear in the number of nodes.

### 3.2 NOTATION

**Spokes and Hubs.** Throughout this paper, we refer to the graph nodes as "spokes" [1] and the added virtual nodes as "hubs". The number of spokes is $N_s$, and they are indexed with $i_s = 1, \ldots, N_s$. Each spoke has features represented by $\boldsymbol{s}_{i_s} \in \mathbb{R}^d$, with the collection of all spoke features denoted by $\boldsymbol{s}$. Similarly, the number of hubs is denoted by $N_h$, and they are indexed by $i_h = 1, \ldots, N_h$. Each hub has features represented by $\boldsymbol{h}_{i_h} \in \mathbb{R}^d$, and the collection of all hub features is denoted by $\boldsymbol{h}$. The binary matrix $\boldsymbol{E} \in \{0, 1\}^{N_s \times N_h}$, referred to as the *hub assignment matrix*, indicates which spokes are connected to which hubs:

$$\boldsymbol{E}_{i_s, i_h} = \begin{cases} 1 & \text{if spoke } i_s \text{ is connected to hub } i_h \\ 0 & \text{otherwise} \end{cases} \quad \text{s.t.} \quad \boldsymbol{E}\,\boldsymbol{1}_{N_h} = k \cdot \boldsymbol{1}_{N_s}, \quad (1)$$

where $\boldsymbol{1}_N$ denotes an all-ones column vector of length $N$. Namely, a given spoke is connected to exactly $k$ hubs, where $k = O(1)$. $\boldsymbol{E}$ can be implemented as a sparse assignment matrix. Finally, where relevant the network layer is denoted using a superscript.

---

[1]In this work, we use "spokes" to represent graph nodes, deviating from the more common usage where "spokes" refer to edges.

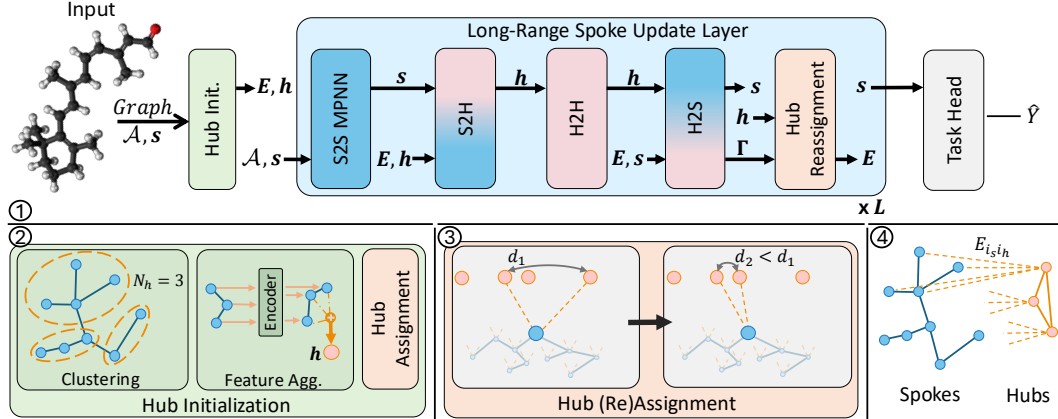

Figure 1: Illustration of ReHub architecture. (1) Overview of the different steps in the architecture. $\mathcal{A}$ is the input (spoke) graph's adjacency matrix; spoke features $s$; hub features $h$; hub assignment $E$; and $\Gamma$ contains attention scores from the Hubs-Spokes attention. (2) Hubs initialization. Spokes are first clustered, then each cluster is aggregated to compute hub features. Finally, each spoke is assigned more hubs. (3) Hub (Re)Assignment. A spoke connected to $k$ hubs updates its hub assignment to the $k$ hubs closest to the hub most similar to it. (4) Illustration of connectivity between spokes and hubs. Information pass between spokes with an MPNN; while the interaction between hubs and spokes is performed via attention and only through available connections $E_{i_s i_h}$. The hubs pass information between themselves via full self-attention.

**Bipartite graph attention.** In our pipeline we utilize graph attention (Brody et al., 2021) between spokes and hubs that interchangeably act as source and target graphs. The definition is as follows:

$$O\,,\,\Gamma = \texttt{Attention}\left(K, Q, E\right), \qquad (2)$$

where the input consists of source graph nodes $K \in \mathbb{R}^{n_k \times d}$, destination graph nodes $Q \in \mathbb{R}^{n_q \times d}$, and a hub assignment $E \in \{0, 1\}^{n_k \times n_q}$ containing the connections between the nodes of the source graph with those of the destination graph. The outputs are the per-node updated features $O \in \mathbb{R}^{n_q \times d}$ of the destination graph. We optionally output the sparse attention scores $\Gamma \in \mathbb{R}^{n_k \times n_q}$, computed for the non-zero entries of $E$. For a non-bipartite graph the same formalism can be applied by taking $K = Q$, enabling self-attention within the graph.

### 3.3 INITIALIZATION

We initiate ReHub by creating $N_h = r\sqrt{N_s}$ hubs, where $r$ is the hub-ratio, set to 1 in almost all benchmarks. To populate the hubs features with meaningful values we compute them based on (i) clustering the input graph (*i.e.* the graph of spokes), specified by the adjacency matrix $\mathcal{A}$, and (ii) aggregating the spoke features $s$. We then assign each spoke to $k$ hubs. This process is illustrated in module (2) of Figure 1.

**Clustering.** We partition the graph $\mathcal{A}$, along with its spoke features $s$, into $N_h$ clusters using METIS (Karypis & Kumar, 1998). This method takes as input an adjacency matrix between spokes $\mathcal{A}$ as well as the desired number of clusters $N_h$; and returns as output a cluster index for each spoke. We denote each cluster as $\mathcal{C}_{i_h}$, where a spoke belongs to a cluster if $i_s \in \mathcal{C}_{i_h}$.

**Hub features.** For each cluster $\mathcal{C}_{i_h}$, we compute the initial hub features. For hub-cluster $i_h$, we aggregate the spoke features as follows:

$$h_{i_h}^0 = \texttt{Aggregate-Feat}\left(\{s_{i_s}^0\}_{i_s \in \mathcal{C}_{i_h}}\right) \qquad (3)$$

There are several options for the `Aggregate-Feat` function. For categorical or ordinal variables (*e.g.*, , atom type), one may compute a histogram. For continuous variables, one may compute the average. In our case, we choose to average the features after the nodes have passed through a positional encoding layer followed by a feedforward layer.

In Section 4.3, we compare our initialization scheme with those proposed in previous works and demonstrate that it significantly improves performance.

**Hub assignment.** We aim to connect each spoke with a total of $k$ hubs. Since each spoke $s_{i_s}$, already has a single connection to a hub $h_{i_h}$ as a result of the clustering, we assign the remaining $k - 1$ hub connections for each spoke, by selecting the hubs closest to $h_{i_h}$ in terms of feature similarity. The assignment procedure is detailed in Section 3.4. This step results in the spoke-to-hub initial assignment matrix $E_{s \to h}^0$.

## 3.4 LONG-RANGE SPOKE UPDATE LAYER

A key component of ReHub's architecture is the **long-range spoke update layer**, which leverages both the spoke graph and the connections between spokes and hubs. Specifically, to maintain efficiency, we avoid global spoke-to-spoke attention operations by using local message passing and global spoke-to-hub operations. We further keep the spoke-to-hub attention efficient by restricting the hub connectivity of spokes to a small constant number of hubs, $k$. Long-range information flow between spokes occurs through hub-to-hub communication, which is made efficient by selecting a number of hubs proportional to the square root of the number of spokes. This relaxes the restriction imposed by previous graph transformer methods (Shirzad et al., 2023), which limited the number of hubs to a small constant. This layer is repeated $L$ times throughout the network. Each layer $\ell$ consists of five steps, as shown in Figure 1, which we describe in detail below.

**(1) Spokes $\to$ Spokes:** For the local spoke update using the neighboring nodes in the graph:

$$s^{\ell + \frac{1}{2}} = \text{MPNN}(s^{\ell}), \tag{4}$$

where MPNN is a single layer of a message-passing neural network. Since this operation is restricted to the 1-ring neighborhood, it remains efficient. Notably, any type of message-passing scheme can generally be integrated into our pipeline.

**(2) Spokes $\to$ Hubs:** Given the updated spokes, we update the hubs using the assignment matrix. Since each spoke is connected to exactly $k$ hubs, the operation is sparse and memory efficient.

$$h^{\ell + \frac{1}{2}} = \text{Attention}(s^{\ell + \frac{1}{2}}, h^{\ell}, E_{s \to h}^{\ell}) \tag{5}$$

**(3) Hubs $\to$ Hubs:** The hubs now interact using self-attention. Even though the connectivity $E_{full}$ is dense, the overall number of hubs is kept on the order of $\sqrt{N_s}$, ensuring overall linear complexity.

$$h^{\ell + 1} = \text{Attention}(h^{\ell + \frac{1}{2}}, h^{\ell + \frac{1}{2}}, E_{full}) \tag{6}$$

**(4) Hubs $\to$ Spokes:** Given the hubs, we update the spokes:

$$s^{\ell + 1}, \Gamma^{\ell + 1} = \text{Attention}(h^{\ell + 1}, s^{\ell + \frac{1}{2}}, E_{h \to s}^{\ell}) \qquad \Gamma^{\ell + 1} \text{ is } N_s \times N_h \tag{7}$$

where the matrix $E_{h \to s}^{\ell} = \left(E_{s \to h}^{\ell}\right)^T$, assuring the same efficiency as the Spokes $\to$ Hubs step.

**(5) Hub (Re)Assignment:** While restricting each spoke to connect with $k$ hubs maintains efficiency, for the method to achieve its full potential, it should utilize all available hubs. We achieve this by keeping only $k$ connected hubs per spoke at each layer, while allowing each spoke to reassign $k - 1$ of its hubs before proceeding to the next layer[2]. The reassignment is based on spoke-hub similarity. Selecting the hubs closest to each spoke, however, would require $N_s \times N_h$ computations. To avoid this, we leverage the distances between hubs, which are efficient to compute.

We then retain the hub most similar to each spoke from the sparse set of connected hubs, replacing the remaining $k - 1$ hubs with those closest to it. This procedure is outlined in Algorithm 1. The matrix of distances between all hub features is denoted by $\Delta$. Naturally, a hub is closest to itself and would be selected first. Additionally, we use $\Gamma$, the attention score matrix, to identify, for each spoke, the most similar hub to which it is connected.

---

[2]We note that, although we replace $k - 1$ hubs while keeping the closest hub connected, in later layers, that closest hub may change. This flexibility prevents us from being overly constrained by the initial hub selection.

---

**Algorithm 1** Hub (Re)Assignment

---

**Require:** Hub-Spoke cross-attention score matrix $\mathbf{\Gamma}^{\ell+1}$ and Hub-Hub distance matrix $\mathbf{\Delta}^{\ell+1}$

    **for** $i_h = 1$ to $N_h$ **do**
        $\mathcal{H}(i_h) = \texttt{Bottom-k-Indices}(\text{row } i_h \text{ of } \mathbf{\Delta}^{\ell+1})$
    **end for**
    **for** $i_s = 1$ to $N_s$ **do**
        $i_h^* = \arg\max_{i_h} \mathbf{\Gamma}^{\ell+1}_{i_s i_h}$
        $\boldsymbol{E}^{\ell+1}_{i_s i_h} = \begin{cases} 1 & \text{if } i_h \in \mathcal{H}(i_h^*) \\ 0 & \text{otherwise} \end{cases}$
    **end for**
    **return** $\boldsymbol{E}^{\ell+1}$

---

**Final prediction.** The pipeline concludes with a task-specific prediction head. In this work, we demonstrate tasks such as graph classification, regression, node classification, and link prediction. These prediction heads use an MLP on the final spoke feature predictions, $\boldsymbol{s}^L$, with further aggregation for graph-level tasks. For link prediction, the pipeline computes a similarity score for every pair of nodes connected by an edge.

**Complexity** Recall that sparse attention is used, where multiplications are performed only between spokes and the $k$ hubs to which they are connected. The resulting time and memory complexity for each Spokes-to-Hubs interaction step is $O(N_s k)$, and for the Hubs self-attention step, it is $O(N_h^2)$. By taking $N_h = O(\sqrt{N_s})$ and $k = O(1)$, we achieve linear complexity in the total number of spokes $N_s$, as desired.

## 4 EXPERIMENTS

**Methods in comparison.** We compare ReHub against leading graph transformer based methods. GraphGPS (Rampášek et al., 2022) offers a framework to integrate MPNNs of different types (Kipf & Welling, 2016; Chen et al., 2020; Hu et al., 2019; Bresson & Laurent, 2017) and transformers. Transformer indicates a straightforward adaptation of the standard transformer architecture Vaswani et al. (2017) to graphs. Spectral Attention Networks (SAN) (Kreuzer et al., 2021) employ attention on the fully connected graph in addition to graph attention using the original edges. Closest to our method is Neural Atoms (Li et al., 2024) which utilizes a set of different, learned, virtual nodes at each layer. Neural Atoms is able to propagate long-range information, improve performance across various tasks and can be modularly applied to different MPNNs. As opposed to Neural Atoms, in this work we aim to tackle the efficiency aspect, *i.e.* to maintain linear complexity in the number of nodes in a graph without a loss of performance. Exphormer (Shirzad et al., 2023) introduces a transformer architecture that achieves linear computational complexity by leveraging expander graphs (Alon, 1986) to define sparse attention patterns. In this model, each node attends only to a fixed number of neighbors specified by a fixed expander graph, and a few global virtual nodes connected to all nodes are used to capture global context. In contrast, our proposed method ReHub employs a dynamic model where virtual nodes (hubs) are connected to subsets of nodes (spokes) rather than to all nodes. ReHub allows for rewiring connections between layers, enhancing adaptability while avoiding bottlenecks associated with fully connected global nodes, all while maintaining linear computational complexity.

**Datasets.** We evaluate ReHub on (1) long-range communication ability and (2) large graphs to verify memory efficiency. For long-range communication we evaluate ReHub on the long-range graph benchmark (LRGB) which is is widely used to evaluate methods which aim at overcoming issues such as oversmoothing and oversquashing. LRGB comprises five datasets. Two of the datasets are image-based graph datasets: PascalVOC-SP and COCO-SP which contain superpixel graphs of the well known image segmentation datasets PascalVOC and COCO. The latter three datasets are molecular datasets: Peptides-Func, Peptides-Struct and PCQM-Contact, which require the prediction of molecular interactions and properties that require global aggregation of information. For evaluation on large graphs we show results on graph datasets of citation networks: OGBN-Arxiv and Coauthor Physics which include about 170K and 30K nodes respectively, with the task of node class prediction. Additionally, we evaluate peak memory consumption on a custom dataset of large

Table 1: **MPNN modularity**. Test performance on datasets from the long-range graph benchmarks (LRGB) (Dwivedi et al., 2022) compared on various GNN types to Neural Atoms (Li et al., 2024). ReHub-FC has each spoke fully connected to all hubs. Best results are colored: first, second.

| Model | Peptides-func | Peptides-struct | PCQM-Contact |
|---|---|---|---|
| | AP ↑ | MAE ↓ | MRR ↑ |
| GCN | 0.5930 ± 0.0023 | 0.3496 ± 0.0013 | 0.2329 ± 0.0009 |
| + NeuralAtoms | 0.6220 ± 0.0046 | 0.2606 ± 0.0027 | 0.2534 ± 0.0200 |
| **+ ReHub-FC** | **0.6663 ± 0.0053** | **0.2489 ± 0.0011** | **0.3492 ± 0.0012** |
| **+ ReHub** | **0.6656 ± 0.0043** | **0.2497 ± 0.0021** | **0.3469 ± 0.0014** |
| GCN2 | 0.5543 ± 0.0078 | 0.3471 ± 0.0010 | 0.3161 ± 0.0004 |
| + NeuralAtoms | 0.5996 ± 0.0033 | 0.2563 ± 0.0020 | 0.3049 ± 0.0006 |
| **+ ReHub-FC** | **0.6427 ± 0.0085** | **0.2511 ± 0.0015** | **0.3386 ± 0.0026** |
| **+ ReHub** | **0.6406 ± 0.0030** | **0.2530 ± 0.0029** | **0.3375 ± 0.0013** |
| GINE | 0.5498 ± 0.0079 | 0.3547 ± 0.0045 | 0.3180 ± 0.0027 |
| + NeuralAtoms | 0.6154 ± 0.0157 | 0.2553 ± 0.0005 | 0.3126 ± 0.0021 |
| **+ ReHub-FC** | **0.6682 ± 0.0098** | **0.2506 ± 0.0012** | **0.3426 ± 0.0014** |
| **+ ReHub** | **0.6582 ± 0.0095** | **0.2514 ± 0.0056** | **0.3429 ± 0.0014** |
| GatedGCN | 0.5864 ± 0.0077 | 0.3420 ± 0.0013 | 0.3218 ± 0.0011 |
| + NeuralAtoms | 0.6562 ± 0.0075 | 0.2585 ± 0.0017 | 0.3258 ± 0.0003 |
| **+ ReHub-FC** | **0.6732 ± 0.0107** | **0.2501 ± 0.0034** | **0.3526 ± 0.0014** |
| **+ ReHub** | **0.6685 ± 0.0074** | **0.2512 ± 0.0018** | **0.3534 ± 0.0014** |
| GatedGCN+RWSE | 0.6069 ± 0.0035 | 0.3357 ± 0.0006 | 0.3242 ± 0.0008 |
| + NeuralAtoms | 0.6591 ± 0.0050 | 0.2568 ± 0.0005 | 0.3262 ± 0.0010 |
| **+ ReHub-FC** | **0.6690 ± 0.0025** | **0.2490 ± 0.0075** | **0.3523 ± 0.0012** |
| **+ ReHub** | **0.6653 ± 0.0054** | **0.2488 ± 0.0017** | **0.3528 ± 0.0008** |

random regular graphs (Steger & Wormald, 1999; Kim & Vu, 2003) of gradually increasing sizes from 10K to 700K nodes. Additional statistics about the datasets is available in Appendix A.1

**Metrics and evaluation.** We evaluate our models using several metrics: Average Precision (AP), Mean Absolute Error (MAE), Mean Reciprocal Rank (MRR), F1 Score, and Accuracy. These are standard metrics and we refer to Dwivedi et al. (2022) for more details. For the evaluation of ReHub, we report the mean ± std over 5 runs, each with a different random seed.

**Hardware.** All experiments were performed using one NVIDIA L40 GPU with 48GB of memory.

### 4.1 LONG-RANGE GRAPH BENCHMARK

A major challenge in graph learning is long range communication – scenarios where the prediction relies on information residing at far location of the graph. In this experiment, we evaluate ReHub on a set of such tasks provided by the LRGB dataset. We split the comparison in two, first establishing the modularity of ReHub by integrating it with various MPNN layers; we then follow with a comparison against leading methods.

**MPNN modularity.** Similar to Neural Atoms, which improves performance across various MPNNs, ReHub is equally modular. In Table 1, we present a comparison of ReHub using several common MPNNs. Results are shown for both the sparse case—where the number of hubs connected to each spoke per layer, $k$, is small—and a dense variant, ReHub-FC, where each spoke is fully connected to all hubs. The performance of both the sparse and dense configurations is compared to Neural Atoms as well the vanilla MPNN technique, demonstrating significant improvements across datasets and MPNNs. Interestingly, we observe that the performance of Neural Atoms is strongly affected by the base MPNN used. *e.g.*, for Peptides-func the final AP ranges between 0.60 and 0.66, while ReHub demonstrates increased robustness ranging between 0.64 and 0.67. Remarkably, thanks to our reassignment procedure, which promotes high utilization of all hubs, our sparse version achieves performance comparable to the dense version.

Table 2: Test performance on datasets from the long-range graph benchmarks (LRGB) (Dwivedi et al., 2022) compared to baselines. For Neural Atoms we show only available results. ReHub-FC has each spoke fully connected to all hubs. Best results are colored: **first**, **second**.

| Model | Peptides-func | Peptides-struct | PCQM-Contact | PascalVOC-SP |
|---|---|---|---|---|
| | AP $\uparrow$ | MAE $\downarrow$ | MRR $\uparrow$ | F1 Score $\uparrow$ |
| Transformer+LapPE | $0.6326 \pm 0.0126$ | $0.2529 \pm 0.0016$ | $0.3174 \pm 0.0020$ | $0.2694 \pm 0.0098$ |
| SAN+LapPE | $0.6384 \pm 0.0121$ | $0.2683 \pm 0.0043$ | $0.3350 \pm 0.0003$ | $0.3230 \pm 0.0039$ |
| GraphGPS | $0.6535 \pm 0.0041$ | $0.2500 \pm 0.0005$ | $0.3337 \pm 0.0006$ | $0.3748 \pm 0.0109$ |
| Exphormer | $0.6527 \pm 0.0043$ | $0.2481 \pm 0.0007$ | $0.3637 \pm 0.0020$ | $0.3975 \pm 0.0037$ |
| NeuralAtoms | $0.6591 \pm 0.0050$ | $0.2553 \pm 0.0005$ | $0.3262 \pm 0.0010$ | n/a |
| ReHub-FC (Ours) | $0.6732 \pm 0.0107$ | $0.2489 \pm 0.0011$ | $0.3526 \pm 0.0014$ | $0.3526 \pm 0.0045$ |
| ReHub (Ours) | $0.6685 \pm 0.0074$ | $0.2488 \pm 0.0017$ | $0.3534 \pm 0.0014$ | $0.3860 \pm 0.0172$ |

Table 3: Coauthor Physics (Shchur et al., 2018) and OGBN-Arxiv (Hu et al., 2020) test results show ReHub achieves comparable accuracy to Exphormer with significant reduction in memory consumption.

| Model | Coauthor Physics | | OGBN-Arxiv | |
|---|---|---|---|---|
| | Peak Memory (GB) $\downarrow$ | Accuracy $\uparrow$ | Peak Memory (GB) $\downarrow$ | Accuracy $\uparrow$ |
| GraphGPS (Transformer) | OOM | - | OOM | - |
| Exphormer | 1.77 | $97.16 \pm 0.13$ | 2.83 | $72.44 \pm 0.28$ |
| ReHub (Ours) | **1.13** | $96.89 \pm 0.19$ | **2.45** | $71.06 \pm 0.40$ |

**Comparison with baselines.** In Table 2 we present a comparison between our ReHub using the best performing MPNN and state of the art methods on the LRGB benchmark. As can be seen, ReHub consistently scores among the top two methods.

### 4.2 PERFORMANCE ON LARGE GRAPHS

**Peak memory vs. graph size.** Our method demonstrates linear memory complexity. To showcase this in practice, we compare the peak memory consumption of our approach to other methods on graphs of varying sizes. Since no existing benchmark offers a collection of gradually growing graph sizes we instead construct a series of toy graphs with sizes varying between 10K and 700K. The graphs are $d$-regular (*i.e.* each node has $d$ neighbors), and are populated with random node features and edge attributes. In this experiment, we set $d = 3$. To keep the comparison fair, for all methods we used similar parameters like the number of layers and hidden dimension. A detailed description of the used parameters is provided in Appendix A.2. For Neural Atoms, we follow the guidelines provided in the paper and use a ratio of 0.1 for the number of virtual nodes. As this ratio results in an asymptotic complexity of $O(N_s^2)$ we additionally include results for Neural Atoms with $N_h = \sqrt{N_s}$ for a more memory efficient version reaching $O(N_s^{3/2})$. For Exphormer, the expander graph has a degree of 3, which is the same value as $k$ used for ReHub. The results, shown in Figure 2, indicate that our method uses less than half the memory of other methods while exhibiting a linear memory usage trend.

**Memory consumption and accuracy on large graphs benchmarks.** We evaluate ReHub on the competitive Coauthor Physics and OGBN-Arxiv datasets which have about 35K and 170K nodes respectively. Comparing ReHub to Exphormer on Table 3 shows an improved memory consumption by about 36% for Coauthor Physics and 13% for OGBN-Arxiv while accuracy is comparable. We follow Exphormer and include GraphGPS (with vanila Transformer) in the comparison to highlight that these graph sizes are considered challenging to process.

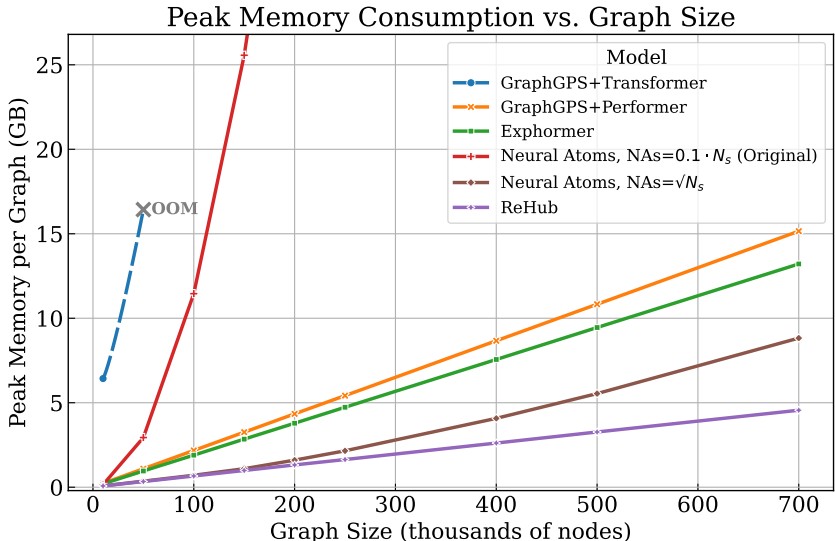

Figure 2: **Peak memory consumption for different architectures**. We compare ReHub to Neural Atoms (Li et al., 2024) and other architectures, and show that ReHub memory consumption is both linear in the number of nodes and requires less memory.

### 4.3 ABLATIONS AND ANALYSES

**Long-range spoke update layer components.** As described in Section 3, our long-range spoke update layer is designed to handle long range communication while maintaining memory efficiency. The ablation provided in Table 4 highlights the contributions of the primary design choices. (1) Initialization of hub features from spokes vs. learned hub features as parameters, where the latter is analogous to the initialization process of Neural Atoms. The initialization scheme, outlined in Section 3.3, is compared to a configuration where hub features are learned parameters, i.e, $\boldsymbol{h}_{i_h}^0 = \boldsymbol{P}_{i_h} \in \mathbb{R}^d$. Initializing the hubs from spokes significantly improves performance, while using learned hubs results in reduced performance compared to the GNN baseline. (2) The use of a fixed vs. dynamic number of hubs. For the fixed case, we set the number of hubs to 22, which is approximately the square root of the average number of nodes, $\sqrt{479.4} \approx 21.9$. This is compared to setting the number of hubs as $N_h = \sqrt{N_s}$ dynamically, according to graph size. (3) Reassigning spokes to hubs at each layer vs. keeping the same connections fixed across the layers, is shown to improve performance. (4) Including an encoding layer for the spokes prior to aggregation further improves performance.

We provide additional experiments for varying values of hubs ratios and k in Appendix A.4.

**Hub utilization.**

To gain insights into the reassignment dynamics, we measure the level of hub utilization. We define hub utilization $U$ as the number of hubs that have at least one spoke connected to them. Formally, $U = |\{i_h \mid \boldsymbol{E}_{:,i_h} \cdot \mathbf{1}_{N_h} \geq 1\}|$ where $\boldsymbol{E}_{:,i_h}$ represents the set of all spokes connected to hub $i_h$. The percentage of unused hubs is then given by $1 - U/N_h$, which reflects the proportion of hubs that are not connected to any spoke. Figure 3 presents a cumulative graph illustrating the percentage of graphs with an unused hub percentage below a given threshold. Hub utilization is shown per layer for different configurations of hub ratio $r$ and connected hubs $k$, based on the validation split of the PascalVOC-SP dataset. The results consistently demonstrate that approximately 10% of hubs remain isolated in each graph, across layers, regardless of the number of hubs or the number of spokes per hub. This finding suggests that the network maintains robust information flow between spokes and hubs. We also present an additional metric based on the Bhattacharyya coefficient in Appendix A.4.

Table 4: **Ablation study.** We measure the effect of various components of ReHub on top of a GatedGCN MPNN, using the PascalVOC-SP dataset. The number of hubs used per graph (#Hubs): for 22 it is a static amount and for $\sqrt{N_s}$ it is dynamic per graph size. Initial hubs (Hubs Init) can be set as learned parameters or initialized from the assigned spokes as described in 3.3 where we can add a feedforward layer on the spokes (Spokes Enc) before aggregation. Reassignment is as described in 3.4. We use $k = 3$ for all runs.

| GNN | #Hubs | Hubs Init | Spokes Enc | Reassignment | PascalVOC-SP (F1 ↑) |
|---|---|---|---|---|---|
| + | - | - | - | - | $0.3152 \pm 0.0045$ |
| + | 22 | Learned (As in Neural Atoms) | - | - | $0.3084 \pm 0.0044$ |
| + | 22 | Cluster Mean | - | - | $0.3574 \pm 0.0065$ |
| + | $\sqrt{N_s}$ | Cluster Mean | - | - | $0.3703 \pm 0.0086$ |
| + | $\sqrt{N_s}$ | Cluster Mean | - | + | $0.3797 \pm 0.0123$ |
| + | $\sqrt{N_s}$ | Cluster Mean | + | - | $0.3775 \pm 0.0040$ |
| + | $\sqrt{N_s}$ | Cluster Mean | + | + | $0.3860 \pm 0.0172$ |

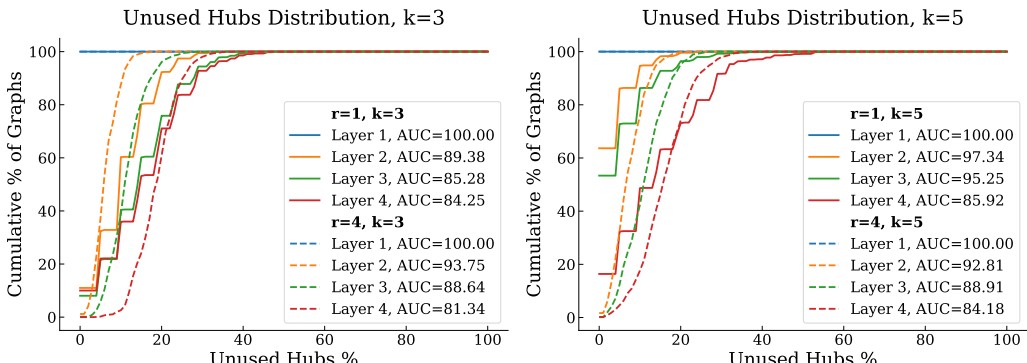

Figure 3: **Hub utilization**. Unused hubs distribution for varying $k$ and hub ratio $r$ evaluated on the validation split of PascalVOC-SP dataset. Left: $k = 3$ with $r \in \{1, 4\}$. Right: $k = 5$ with $r \in \{1, 4\}$.

## 5 CONCLUSION AND FUTURE WORK

In this paper, we introduced ReHub, a novel graph transformer architecture designed to enhance long-range communication in large graphs through dynamic reassignment of virtual nodes. This approach facilitates efficient information passage, enabling effective learning on large graphs while maintaining linear computational complexity and reduced memory usage compared to existing methods. Our experimental results demonstrate that ReHub consistently outperforms Neural Atoms and ranks among the top two methods against various baselines. It achieves competitive accuracy compared to Exphormer with lower memory consumption across all evaluated scenarios. ReHub achieves its efficiency through our proposed reassignment mechanism, which maintains sparse spoke-hub connectivity, as highlighted by our ablation studies. The modular design of ReHub allows for easy integration with various MPNNs.

**Limitations** While consistently improving computational efficiency, our spoke-hub communication maintained strong performance, though it did not establish a new state of the art. Additionally, our current design lacks inherent support for positional information available in geometric graphs. In future work, we aim to extend our method to support tasks requiring long-range communication on geometric graphs by incorporating positional information into the spoke-hub attention and reassignment mechanisms. We also plan to make the reassignment module learnable, and optimize it towards the prediction task to further boost accuracy. Finally, integrating ReHub into architectures like Exphormer to enable the reassignment of expander graph edges between layers presents a promising direction to further enhance long-range communication.

REPRODUCIBILITY

An overview of the hyperparamters used for training in any of the settings can be found in Appendix A.2 and a description of the experimental setup used for training and evaluation in Section 4. Additionally, we make our code publicly available: https://anonymous.4open.science/r/ReHub-C366/

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

# A  APPENDIX

## A.1  DATASETS

In Tables 5 and 6 we summaries the details of the datasets used for evaluation.

Table 5: Statistics of the five dataset proposed in the long-range graph benchmark.  Source: LRGB (Dwivedi et al., 2022).

| Dataset | Total Graphs | Total Nodes | Avg Nodes | Mean Deg. | Total Edges | Avg Edges | Avg Short.Path. | Avg Diameter |
|---|---|---|---|---|---|---|---|---|
| PascalVOC-SP | 11,355 | 5,443,545 | 479.40 | 5.65 | 30,777,444 | 2,710.48 | 10.74±0.51 | 27.62±2.13 |
| COCO-SP | 123,286 | 58,793,216 | 476.88 | 5.65 | 332,091,902 | 2,693.67 | 10.66±0.55 | 27.39±2.14 |
| PCQM-Contact | 529,434 | 15,955,687 | 30.14 | 2.03 | 32,341,644 | 61.09 | 4.63±0.63 | 9.86±1.79 |
| Peptides-func | 15,535 | 2,344,859 | 150.94 | 2.04 | 4,773,974 | 307.30 | 20.89±9.79 | 56.99±28.72 |
| Peptides-struct | 15,535 | 2,344,859 | 150.94 | 2.04 | 4,773,974 | 307.30 | 20.89±9.79 | 56.99±28.72 |

Table 6: Dataset statistics of LRGB, OGBN-Arxiv and Coauthor Physics.  Source: Exphormer (Shirzad et al., 2023)

| Dataset | Graphs | Avg. nodes | Avg. edges | Prediction Level | No. Classes | Metric |
|---|---|---|---|---|---|---|
| PascalVOC-SP | 11,355 | 479.4 | 2,710.5 | inductive node | 21 | F1 |
| COCO-SP | 123,286 | 476.9 | 2,693.7 | inductive node | 81 | F1 |
| PCQM-Contact | 529,434 | 30.1 | 61.0 | inductive link | (link ranking) | MRR |
| Peptides-func | 15,535 | 150.9 | 307.3 | graph | 10 | Average Precision |
| Peptides-struct | 15,535 | 150.9 | 307.3 | graph | 11 (regression) | Mean Absolute Error |
| OGBN-Arxiv | 1 | 169,343 | 1,166,243 | node | 40 | Accuracy |
| Coauthor Physics | 1 | 34493 | 247962 | node | 5 | Accuracy |

## A.2 Hyperparameters

**Long-range graph benchmark.** In the experiments done on the datasets: Peptides-func, Peptides-struct and PCQM-Contact we follow the hyperparameters of Neural Atoms (Li et al., 2024). For other datasets we follow the hyperparameters of Exphormer (Shirzad et al., 2023). Although we follow the hyperparameters configurations, there may be subtle changes due to the difference in architecture.

In Tables 7- 10 we provide the hyperparameters used in our experiments.

Table 7: Hyperparameters for the LRGB datasets used for evaluation. For Peptides-func, Peptides-struct and PCQM-Contact some of the hyperparameters are model-specific and presented in additional tables.

| Hyperparameter | PCQM-Contact | Peptides-func | Peptides-struct | PascalVOC-SP |
|---|---|---|---|---|
| Dropout | 0 | 0.12 | 0.2 | 0.15 |
| Attention dropout | 0.2 | 0.2 | 0.2 | 0.2 |
| Positional Encoding | LapPE-10 | LapPE-10 | LapPE-10 | LapPE-10 |
| PE Dim | 16 | 16 | 20 | 16 |
| PE Layers | 2 | 2 | 2 | 2 |
| PE Encoder | DeepSet | DeepSet | DeepSet | DeepSet |
| Batch size | 256 | 128 | 128 | 32 |
| Learning Rate | 0.0003 | 0.0003 | 0.0003 | 0.0005 |
| Weight Decay | 0 | 0 | 0 | 0 |
| Warmup Epochs | 10 | 10 | 10 | 10 |
| Optimizer | AdamW | AdamW | AdamW | AdamW |
| # Epochs | 200 | 200 | 200 | 300 |
| MPNN | - | - | - | GatedGCN |
| # Layers | - | - | - | 4 |
| Hidden Dim | - | - | - | 96 |
| # Heads | - | - | - | 8 |
| Hubs Ratio | - | - | - | 1 |
| k | - | - | - | 3 |

Table 8: Model-specific hyperparameters for PCQM-Contact.

| Hyperparameter | # Layers | Hidden Dim | # Heads | Hubs Ratio | k |
|---|---|---|---|---|---|
| GCN | 5 | 300 | 1 | 1 | 3 |
| GCNII | 5 | 100 | 2 | 1 | 3 |
| GINE | 5 | 100 | 1 | 1 | 3 |
| GatedGCN | 8 | 72 | 1 | 1 | 3 |

Table 9: Model-specific hyperparameters for Peptides-func.

| Hyperparameter | # Layers | Hidden Dim | # Heads | Hubs Ratio | k |
|---|---|---|---|---|---|
| GCN | 5 | 155 | 1 | 0.5 | 3 |
| GCNII | 5 | 88 | 1 | 0.5 | 3 |
| GINE | 5 | 88 | 2 | 0.5 | 3 |
| GatedGCN | 5 | 88 | 1 | 0.5 | 3 |

Table 10: Model-specific hyperparameters for Peptides-struct.

| Hyperparameter | # Layers | Hidden Dim | # Heads | Hubs Ratio | k |
|---|---|---|---|---|---|
| GCN | 5 | 155 | 1 | 0.5 | 3 |
| GCNII | 5 | 88 | 1 | 0.5 | 3 |
| GINE | 5 | 88 | 2 | 0.5 | 3 |
| GatedGCN | 5 | 88 | 1 | 0.5 | 3 |

**Large random regular graph.** In Table 11 we provide the hyperparameters used in our experiments. The same configuration of hyperparameters is used for all experiments except for model-specific parameters.

Note that although Exphormer does not utilize its expander graph algorithm here the "Add edge index" hyperparameter is enabled and sets the graph edges as the expander edges. Due to the regularity of the graph, *i.e.* having a degree of $d = 3$ for all nodes, the same linear complexity is imposed.

Table 11: Hyperparameters for the forward pass of the large random regular graph dataset for all the models, including the model-specific configuration.

| Hyperparameter | Large Random Regular Graph | Exphormer | ReHub |
|---|---|---|---|
| MPNN | GCN | - | - |
| # Layers | 3 | - | - |
| Hidden Dim | 52 | - | - |
| # Heads | 4 | - | - |
| Add edge index | - | True | - |
| Num Virtual Nodes | - | 4 | - |
| Hubs Ratio | - | - | 1 |
| k | - | - | 3 |

**OGBN-Arxiv and Coauthor Physics.** For OGBN-Arxiv and Coauthor Physics we follow the hyperparameters of Exphormer (Shirzad et al., 2023), adding our configuration of hubs ratio and $k$. In Table 12 we provide the hyperparameters used in our experiments.

Table 12: Hyperparameters for OGBN-Arxiv and Coauthor Physics datasets used for evaluation.

| Hyperparameter | OGBN-Arxiv | Physics |
|---|---|---|
| Dropout | 0.3 | 0.4 |
| Attention dropout | 0.2 | 0.8 |
| Learning Rate | 0.01 | 0.001 |
| Weight Decay | 0.001 | 0.001 |
| Warmup Epochs | 5 | 5 |
| Optimizer | AdamW | AdamW |
| # Epochs | 600 | 70 |
| MPNN | GCN | GCN |
| # Layers | 3 | 4 |
| Hidden Dim | 80 | 72 |
| # Heads | 2 | 4 |
| Hubs Ratio | 1 | 1 |
| k | 3 | 3 |

## A.3 Implementation details

In the following we describe in more detail how the architecture is implemented.

We use the open-source code provided by GraphGPS (Rampášek et al., 2022) and available on https://github.com/rampasek/GraphGPS. We merge into that code parts from Exphormer (Shirzad et al., 2023) which are relevant for the training and evaluation of the OGBN-Arxiv and Coauthor Physics datasets.

**Clustering.** During preprocessing we use the METIS partitioning algorithm (Karypis & Kumar, 1998) to divide each graph to clusters according to the required number of hubs. In practice, we use the python wrapper PyMetis[3] which allows us to map each spoke to a single hub.

**Hub features.** Throughout the implementation we use sparse data structures that allows us to keep the complexity linear even when running on multiple graphs simultaneously (*i.e.* in a batch). The hubs are stored in a similar fashion to how spokes are stored in a batch. *i.e.* the features are kept in a 2D matrix $\boldsymbol{X} \in \mathbb{R}^{N \times d}$ together with a 1D graph index matrix $\boldsymbol{B} \in \{0, \ldots, \#\text{Graphs}\}^N$ indicating which graph each node belongs to, where $N$ is the number of nodes in the whole batch.

When aggregating the spokes to calculate hub features we use the `scatter` functions which allows us to aggregate the spokes to two matrices. (1) a 2D matrix $\boldsymbol{H} \in \mathbb{R}^{H \times d}$ and (2) a 1D graph index matrix $\boldsymbol{B}_H \in \{0, \ldots, \#\text{Graphs}\}^H$ indicating which graph each hub belongs to, where $H$ is the number of hubs in the whole batch.

**Hub assignment and reassignment.** Note that the initial hub assignment is implemented identically to the reassignment algorithm but with $i_h^*$ set to the initial hub found in the clustering step.

**Attention.** Opposed to other transformer architectures which require the conversion of the spokes to a dense representation, we can keep both spokes and hubs in their sparse representation. For each attention module we use `GATv2Conv` implementation provided in pytorch geometric which accept as input this type of sparse representation. Moreover, this implementation accept as input a bipartite graphs and here we use it to pass information between the graph of spokes and graph of hubs.

**Large random regular graph.** As mention in Section 4.2, we would like to construct graphs of arbitrary size. To achieve this, we generate a dataset of large random regular graphs, which can be produced at any scale while ensuring connectivity between nodes (*i.e.* , avoiding isolated subgraphs) due to the regularity property, *i.e.* A d-regular graph is a graph where each node has d number of neighbors. To do that we use the `random_regular_graph` (Kim & Vu, 2003; Steger & Wormald, 1999) function from the open-source `NetworkX` library (Hagberg et al., 2008), which takes as an input the number of nodes to be constructed and the degree of each node $d$. Additionally, to ensure compatibility with the models, we assign random values to the graph's edge attributes, node features, and prediction labels.

**Peak memory usage.** To sample the peak memory usage of the models we use the function `torch.cuda.max_memory_allocated`. This function returns the peak memory allocation since running the `torch.cuda.reset_peak_memory_stats` function, which we call just before the call to the model.

---

[3]https://github.com/inducer/pymetis

### A.4 ADDITIONAL ABLATIONS

**Long-range spoke update layer components.** In addition to the result shown for PascalVOC-SP presented in Section 4.3, we provide in Table 13 the same components analysis for Peptides-func.

Table 13: **Ablation study.** We measure the effect of various components of ReHub on top of a GatedGCN MPNN, using the Peptides-func dataset. The number of hubs used per graph (#Hubs): for 22 it is a static amount and for $\sqrt{N_s}$ it is dynamic per graph size. Initial hubs (Hubs Init) can be set as learned parameters or initialized from the assigned spokes as described in 3.3 where we can add a feedforward layer on the spokes (Spokes Enc) before aggregation. Reassignment is as described in 3.4. We use $k = 3$ for all runs.

| GNN | #Hubs | Hubs Init | Spokes Enc | Reassignment | Peptides-func (AP ↑) |
|---|---|---|---|---|---|
| + | - | - | - | - | $0.5864 \pm 0.0077$ |
| + | 12 | Learned (As in Neural Atoms) | - | - | $0.5738 \pm 0.0027$ |
| + | 12 | Cluster Mean | - | - | $0.6626 \pm 0.0068$ |
| + | $\sqrt{N_s}$ | Cluster Mean | - | - | $0.6616 \pm 0.0063$ |
| + | $\sqrt{N_s}$ | Cluster Mean | - | + | $0.6661 \pm 0.0062$ |
| + | $\sqrt{N_s}$ | Cluster Mean | + | - | $0.6612 \pm 0.0068$ |
| + | $\sqrt{N_s}$ | Cluster Mean | + | + | $0.6683 \pm 0.0069$ |

**Sensitivity to hubs ratio** For ReHub, a practical guideline for selecting the hubs ratio and $k$ is $r = 1$ and $k = 3$. Figure 4 presents an ablation study on these parameters for the Peptides-func and PascalVOC-SP datasets.

For Peptides-func, the best results are achieved with a hubs ratio of $0.5$ for both $k = 3$ and $k = 5$, which may be attributed to the small graph sizes and the potential tendency to overfit on such datasets. On average, each graph contains approximately 150 nodes, resulting in roughly 6 hubs. Furthermore, it is notable that for other hubs ratio values, the results remain consistent, with an average performance between $AP = 0.66$ and $AP = 0.67$, indicating the robustness of our method.

For PascalVOC-SP, which includes larger graphs with an average of approximately 500 nodes per graph, a different trend is observed compared to Peptides-func. Specifically, for varying values of $k$, the optimal performance is achieved with different hubs ratios. However, the performance variation outside the optimal hubs ratio is relatively minor, with the best results obtained when $r = 1$ and $k = 3$.

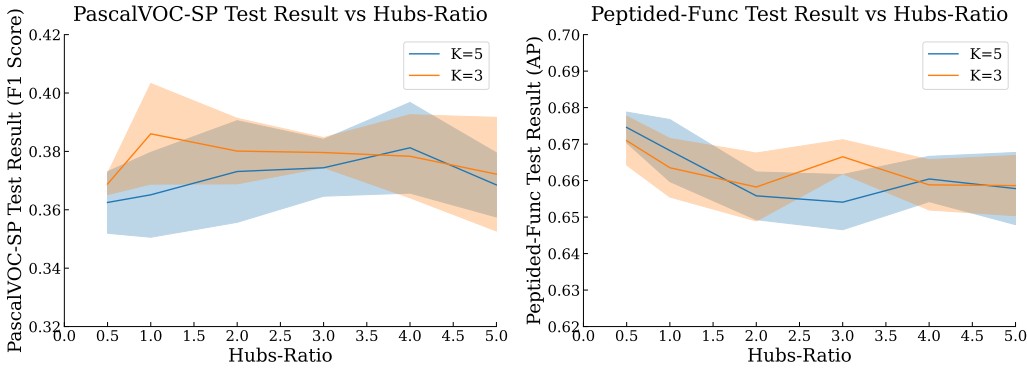

Figure 4: Results for various hubs ratio and k, which is the number of hubs each spoke is connected to. We shown this on PascalVOC-SP (Left) and Peptides-func (Right) datasets with $k = [3, 5]$ and $r = [0.5, 1, 2, 3, 4, 5]$.

**Bhattacharyya Coefficient vs. Uniform Distribution** The Bhattacharyya Coefficient for a discrete probability distributions $P$ and $Q$ is measurement of how similar the two probability distribu-

tions are. It is defined as:

$$BC(P,Q) = \sum_{x \in \mathcal{X}} \sqrt{P(x)Q(x)},$$

$BC(P,Q)$ lies between $0$ and $1$, where the higher the coefficient the more similar the distributions are. In our setup, $P$ is the distribution over spokes connection per hub; *i.e.* it is the number of spokes connected to each hub, divided by the overall number of connections from spokes to hubs (so that the result is indeed a probability distribution). We then set $Q$ to be the uniform distribution, *i.e.* $1/N_h$ for each hub. By doing so, we can see how close the actual distribution $P$ is to the uniform distribution, where uniform indicates the optimal balanced assignment of spokes to hub – where every hub has exactly the same number of spokes. For convenience, we define the Bhattacharyya Percentage as the Bhattacharyya Coefficient multiplied by $100$.

In Figure 5 we present a graph illustrating the percentage of graphs with a Bhattacharyya Percentage below a given threshold for the PascalVOC-SP dataset. As in Section 4.3 this is shown for each layer, and for varying values of hubs ratios and k on the validation split of the dataset. The results demonstrate that regardless of the number of hub and number of hubs per spoke, most graphs have a Bhattacharyya Percentage above 80%. This suggests that our reassignment method spreads nodes quite evenly across the various hubs, and does not create high concentration of spokes which remain connected to only few hubs.

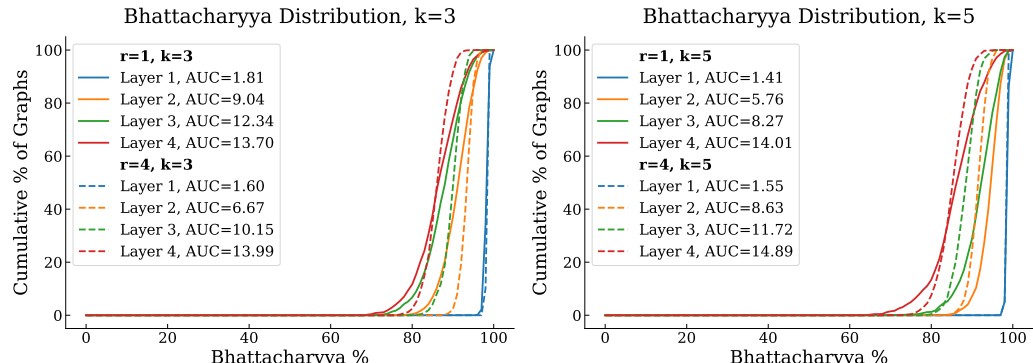

Figure 5: Percentage of graphs with a Bhattacharyya Percentage below a given threshold for the validation split of the PascalVOC-SP dataset. Results are shown for varying $k$ and hubs ratio $r$. Left: $k = 3$ with $r \in \{1, 4\}$. Right: $k = 5$ with $r \in \{1, 4\}$.

