# OpenReview forum: "ReHub: Linear Complexity Graph Transformers with Adaptive Hub-Spoke Reassignment"
_ICLR.cc/2025/Conference — Submitted to ICLR 2025_

### Official Review · Reviewer_XcpA · 2024-11-03

**Soundness:** 2
**Presentation:** 3
**Contribution:** 2
**Rating:** 5
**Confidence:** 3

**Summary:**

GNN is considered an important approach for extensively uncovering and analyzing the intrinsic relational information across various scenarios. However, the rapid expansion of neighborhood space limits its effectiveness. For a long time, many efforts have been made in this area, such as sampling techniques. In the past year or two, the method of introducing virtual nodes has emerged. This paper follows this basic approach and proposes “ReHub,” coining new terms “Spokes” and “Hubs”,  corresponding to graph nodes and virtual nodes, respectively. Furthermore, it dynamically assigns graph nodes to the corresponding virtual nodes. Drawing an analogy to a transportation hub system, for each graph node (termed a “Spoke” in the paper), a fixed-size, interconnected set of virtual nodes, termed a “Hub” by the authors, is assigned. To effectively manage the allocation and reduce overhead, an adaptive reallocation mechanism is employed, as well as hub-to-hub similarity. Overall, this work is seen as transforming one complex problem into another.

**Strengths:**

1. The method is intuitive and easy to understand. It leverages the general observation that each node only needs to interact with a small subset of hubs, enabling the model to maintain linear complexity even as the total number of hubs increases significantly.

2. Accordingly, a key component of ReHub’s architecture is the long-range spoke update layer, which utilizes both the spoke graph and the connections between spokes and hubs.

3. This structure achieves near-linear complexity, preventing rapid expansion of the state space within the graph structure and thereby reducing memory usage and other computational costs.

**Weaknesses:**

1. Overall, this work is seen as transforming one complex problem into another without addressing a fundamental solution to the original issue. The challenge lies in determining the relationship between a node and a hub, as well as the relationships between hubs that connect sets of nodes. The difficulty is not merely in establishing this two-tier relationship but in effectively defining these associations for different types of graphs. Moreover, this structure could potentially be extended to three layers or even more.

2. The paper initiates ReHub by creating  N_h = r \sqrt{N_s}  hubs, where r, the hub-ratio, is set to 1 in most benchmarks. However, this approach lacks a necessary theoretical foundation, raising concerns about its universality. It appears to be more of an empirical method rather than one grounded in rigorous theoretical principles.

3. The testing workloads are quite targeted, focusing on evaluating ReHub’s (1) long-range communication capabilities and (2) memory efficiency on large graphs. Since graph structures in GNNs vary widely, determining whether a particular graph type is well-suited to this approach may also be an open question.

**Questions:**

1. How does the method proposed in this paper differ from the approach of dividing a graph into clusters and finding a central node within each cluster? Although the initial stage of the method in this paper also utilizes basic clustering for classification.

2. r  and  k  are two important hyperparameters;  r  determines the number of hubs, while  k  dictates the number of connections between spokes and hubs. The appendix also analyzes the impact of these two hyperparameters. The question is whether there is a universal method for determining these hyperparameters.

3. Why is there no comparison with clustering methods and sampling methods in the experimental section? Essentially, these methods also analyze the graph as individual subgraphs (or virtual subgraphs), establishing parameter relationships within and between the subgraphs through training.

---

> ### Author Response · Authors · 2024-11-22
>
> We sincerely thank the reviewer for their insightful feedback and comments.
>
>
> **Q1: Difference to clustering methods**
>
> > How does the method proposed in this paper differ from the approach of dividing a graph into clusters and finding a central node within each cluster? Although the initial stage of the method in this paper also utilizes basic clustering for classification.
>
> > Why is there no comparison with clustering methods and sampling methods in the experimental section? Essentially, these methods also analyze the graph as individual subgraphs (or virtual subgraphs), establishing parameter relationships within and between the subgraphs through training.
>
> **A:** Thank you for your insightful questions. You are correct that the initialization in our method involves a clustering technique. However, as you pointed out, simply identifying the central node in each cluster is not sufficient to address tasks like node classification, where information must be passed between nodes. For long-range communication, it’s crucial that information can flow not just within clusters, but also between them.
>
> In our method, clustering is only used during the initialization step to group related spokes, which helps in creating hubs with meaningful features. This initialization is modular, meaning we can use any clustering method to define the initial spoke groups. After this initialization, the spokes are no longer rigidly attached to the clusters, but are connected to multiple hubs. At subsequent steps, these connections are dynamically adjusted through the hub reassignment component, allowing spokes that were initially connected to the same hub to be reassigned to different hubs based on their features. This dynamic reconfiguration enables more flexible and efficient communication between spokes and hubs.
>
> **Q2: Determining r and k**
>
> > r and k are two important hyperparameters; r determines the number of hubs, while k dictates the number of connections between spokes and hubs. The appendix also analyzes the impact of these two hyperparameters. The question is whether there is a universal method for determining these hyperparameters.
>
> **A:** The best r and k were found through cross-validation. Although there is no universal method to instantly determine the best r and k we show that our method is robust to changes in r and k and choosing r=1 and k=3 scores high results. This is opposed to other methods where choosing the number of virtual node hyperparameter is arbitrary and any choice might be valid.
>
>
> **Q3: The method’s challenge**
>
> > Overall, this work is seen as transforming one complex problem into another without addressing a fundamental solution to the original issue. The challenge lies in determining the relationship between a node and a hub, as well as the relationships between hubs that connect sets of nodes. The difficulty is not merely in establishing this two-tier relationship but in effectively defining these associations for different types of graphs. Moreover, this structure could potentially be extended to three layers or even more.
>
> **A:** Thank you for your comment. It is unclear to us what specific "original complex problem" you are referring to, and how it is transformed into another complex problem. We would appreciate further clarification on this point.
>
> In ReHub, hubs serve as auxiliary nodes that facilitate the passage of information between spokes using an attention mechanism. This structure is analogous to transportation systems where people can travel between cities through various routes. Similarly, there are multiple ways for information to flow between spokes, with hubs acting as intermediaries that pass information through one or two hops. The flexibility of this approach allows for various equally effective solutions. The hub reassignment component enables dynamic connections between spokes and hubs, adjusting the relationships based on spoke features and similarity. This dynamic adjustment ensures efficient communication and prevents bottlenecks.
>
> While the two-layer hub-spoke structure works well for many tasks, exploring the addition of more layers (such as a hierarchy of hubs) is indeed an interesting direction for future work. This could potentially further enhance the model’s ability to handle more complex relationships and improve performance across different types of graphs.

---

> > ### Author Response · Authors · 2024-11-22
> >
> > **Q4: Graphs structure**
> >
> > > The testing workloads are quite targeted, focusing on evaluating ReHub’s (1) long-range communication capabilities and (2) memory efficiency on large graphs. Since graph structures in GNNs vary widely, determining whether a particular graph type is well-suited to this approach may also be an open question.
> >
> > **A:** Thank you for the valuable feedback. The idea of evaluating which types of graphs ReHub excels on is an excellent suggestion. In the Long-Range Graph Benchmark (LRGB), we evaluate our method on a variety of graph types, ranging from molecules to superpixels in image segmentation datasets. This diversity allows us to assess the generalizability of ReHub's long-range communication capabilities and memory efficiency across different domains. However, we acknowledge that further exploration of specific graph types and their suitability for our method would provide deeper insights into its strengths and limitations. We plan to expand on this in future work.
> >
> > To further support this point, let’s consider the average graph diameters of these datasets [Ref1]: Peptides-func/struct (56.99±28.72), PascalVOC-SP (27.62±2.13), and PCQM-Contact (9.86±1.79). As shown in Table 2, our method achieves better results on datasets with larger graph diameters. This observation suggests that ReHub is particularly effective on graphs that require long-range communication.
> >
> > [Ref1] Dwivedi, Vijay Prakash, et al. "Long range graph benchmark." Advances in Neural Information Processing Systems 35 (2022): 22326-22340.

---

> ### Comment · Reviewer_XcpA · 2024-11-27
>
> Thank you for your detailed response. As mentioned in the previous comments, drawing an analogy to a transportation hub system: each graph node, referred to as a “Spoke” in the paper, is assigned an interconnected set of virtual nodes, termed a “Hub” by the authors. At its core, transportation systems are essentially graph problems, with transportation being just one of their mathematical applications. Therefore, I stand by my rating.

---

### Official Review · Reviewer_15hB · 2024-11-03

**Soundness:** 2
**Presentation:** 3
**Contribution:** 2
**Rating:** 5
**Confidence:** 4

**Summary:**

This paper introduces ReHub, a novel graph transformer architecture with linear computational complexity. The architecture leverages an adaptive hub-spoke reassignment strategy. Nodes (spokes) are connected to a subset of virtual nodes (hubs) with a deliberately chosen size to optimize long-range communication and maintain efficiency. Through experiments on LRGB and large graph benchmarks, ReHub demonstrates comparable or better performance than existing methods with reduced memory consumption.

**Strengths:**

1. The method demonstrates wide applicability to various graph tasks with a minor modification of the prediction head in architecture.

2. The evaluation section covers various datasets and baseline methods to provide comprehensive results on enhanced long-range information capturing ability over existing methods.

3. The writing and presentation is clear and easy to follow.

**Weaknesses:**

1. The motivation is intuitive: from the spoke-to-hub transportation model, the motivation of such structure is somewhat insufficient. The Hub Reassignment motivation and strategy are intuitively explained, with little further support.

2. In the evaluation of complexity, only (peak) memory consumption is compared with other models to empirically show the low memory complexity, while the claim is that both time and memory complexity are constrained. The empirical results of time complexity remains to be discussed on real-world datasets.

3. The two large graph benchmarks in experiment section seem to still fall within small to medium-sized graphs according to OGB benchmark [Ref.1]. The peak memory consumption of previous method Exphormer on these datasets is only 2~3 GBs. The comparison of memory consumption on these small datasets cannot truly demonstrate the advancement of ReHub because memory is not a bottleneck on these datasets.

[Ref.1] Hu, Weihua, et al. "Open graph benchmark: Datasets for machine learning on graphs." Advances in neural information processing systems 33 (2020): 22118-22133.

**Questions:**

1. In 3.4 (5) Hub (Re)Assignment, why should every spoke utilize all available hubs? It makes sense in transportation domain to balance the load, but it is not well motivated here.

2. From complexity perspective, only peak memory is evaluated to demonstrate that ReHub is memory efficient. With the multi-layer network consisting of message passing, attention and reassignment, is ReHub also empirically computation efficient compared to other methods?

---

> ### Author Response · Authors · 2024-11-22
>
> We sincerely thank the reviewer for their insightful feedback and comments.
>
> **Q1: Hub (Re)Assignment motivation**
>
> > In 3.4 (5) Hub (Re)Assignment, why should every spoke utilize all available hubs? It makes sense in transportation domain to balance the load, but it is not well motivated here.
>
> **A:** Thank you for your comment. We would like to clarify this point. In line 259, we state that "for the method to achieve its full potential, it should utilize all available hubs." By "utilizing hubs," we mean that each hub should have at least one spoke connected to it. The rationale behind this is based on the need to ensure broad information flow across the graph. Unlike transportation systems where the source and destination are predefined, in long-range communication within graphs, we do not know in advance which nodes (or pairs of nodes) need to exchange information. Therefore, we reassign the spokes to different hubs based on their features and similarity to ensure that information can flow across all parts of the graph. Better hub utilization should also help reduce potential bottlenecks. Further analysis of hub utilization is provided in Section 4.3, which we hope will help clarify this design choice.
>
>
> **Q2: Peak memory evaluation**
>
> > From complexity perspective, only peak memory is evaluated to demonstrate that ReHub is memory efficient. With the multi-layer network consisting of message passing, attention and reassignment, is ReHub also empirically computation efficient compared to other methods?
>
> **A:** The peak memory we report is empirically measured and is the maximal memory consumption during the run of our method across all of its components including message passing, attention and reassignment across all layers.
>
> We show in Figure 2 a graph of peak memory consumption for inference on graphs with varying number of nodes. Although this is the peak memory consumption, we carefully run these models with the same parameters to ensure a fair comparison. Still, this leads to ReHub substantially using less memory at peak compared to other methods.
>
> Peak memory consumption is used for evaluation as this is a major constraint for running an inference on large graphs or multiple graphs on a single gpu.
>
>
> **Q3: Motivation of spokes and hubs and support for hub reassignment**
>
> > The motivation is intuitive: from the spoke-to-hub transportation model, the motivation of such structure is somewhat insufficient. The Hub Reassignment motivation and strategy are intuitively explained, with little further support.
>
> **A:** The motivation behind our spoke-hub structure is inspired by transportation models, where the system must support travel between every pair of locations. However, a direct point-to-point model would be inefficient for such a system, which is why a spoke-hub model is commonly adopted to optimize the flow. We adapt this analogy to graph structures, particularly for tasks that require long-range communication (similar to enabling travel between any two locations). This approach reduces the complexity of graph transformers while still ensuring efficient information flow between nodes, allowing us to handle large graphs more effectively.
>
> While this analogy works well for the spoke-hub structure, it does face limitations when applied to hub reassignment. In transportation, the origin and destination are predefined, so the paths are well-known. However, in graphs, we do not have prior knowledge of which nodes (or spokes) need to exchange information to solve a given task. To address this, we introduce hub reassignment, which allows the model to dynamically adjust connections between spokes and hubs at each layer. This adjustment ensures that the relevant spokes are connected to the same hub, preventing bottlenecks in information flow between nodes. We demonstrate the importance and effectiveness of the hub reassignment component in Table 4, where its impact on performance is clearly shown.

---

> > ### Author Response · Authors · 2024-11-22
> >
> > **Q4: Empirical results of time complexity**
> >
> > > In the evaluation of complexity, only (peak) memory consumption is compared with other models to empirically show the low memory complexity, while the claim is that both time and memory complexity are constrained. The empirical results of time complexity remains to be discussed on real-world datasets.
> >
> > **A:** We have provided demonstrations of ReHub’s efficiency in Table 3, where we show our method requires less memory than Exphormer. Moreover, in Figure 2 we provide the peak memory of different methods on varying numbers of graph size during inference time.
> >
> > Additionally, we extend Table 3 and add the average epoch time during inference. The average epoch time is the wall clock time an epoch ran during our evaluation averaged throughout 5 runs with different seeds. As can be seen ReHub achieves both faster inference and lower memory consumption compared to Exphormer.
> >
> > | Model                  |                    | Coauthor Physics  |              |                    | OGBN-Arxiv        |              |
> > |------------------------|--------------------|-------------------|--------------|--------------------|-------------------|--------------|
> > |                        | Peak Memory (GB) ↓ | Avg. Epoch Time (ms) ↓ | Accuracy ↑   | Peak Memory (GB) ↓ | Avg. Epoch Time (ms) ↓ | Accuracy ↑   |
> > | GraphGPS (Transformer) | OOM                | -                 | -            | OOM                | -                 | -            |
> > | Exphormer              | 1.77               | 373 ± 69                 | 97.16 ± 0.13 | 2.83               | 203 ± 7                 | 72.44 ± 0.28 |
> > | ReHub (Ours)           | 1.13               | 320 ± 24              | 96.89 ± 0.19 | 2.45               | 112 ± 9             | 71.06 ± 0.40 |
> >
> >
> >
> > **Q5: Large graph experiments**
> >
> > > The two large graph benchmarks in experiment section seem to still fall within small to medium-sized graphs according to OGB benchmark [Ref.1]. The peak memory consumption of previous method Exphormer on these datasets is only 2~3 GBs. The comparison of memory consumption on these small datasets cannot truly demonstrate the advancement of ReHub because memory is not a bottleneck on these datasets.
> >
> > **A:** For the evaluation of peak memory consumption we constructed a dataset of synthetic graphs with varying number of nodes. We presented the results in Figure 2. The largest graph we evaluated on consists of 700K nodes and the measured peak memory consumption for each model is as follows: GraphGPS (OOM), Performer (15 GB), Exphormer (13GB), NeuralAtoms (9GB) and ReHub (5GB).
> >
> > Please see our general comment for additional information.

---

### Official Review · Reviewer_ThvY · 2024-11-04

**Soundness:** 2
**Presentation:** 2
**Contribution:** 2
**Rating:** 3
**Confidence:** 5

**Summary:**

ReHub is a scalable graph transformer that achieves linear complexity through adaptive spoke-hub reassignment, drawing inspiration from hub-based systems like airlines. By assigning nodes to a limited number of virtual "hubs" and using efficient hub-hub similarity for reassignment, ReHub maintains global attention without high computational costs. Experiments show it outperforms baselines and maintains sparse model performance comparable to dense architectures on long-range benchmarks.

**Strengths:**

- The paper is clearly written and easy to follow.
- The proposed architecture is well-motivated.
- Spokes and Hub is a plausible algorithm for graph learning.

**Weaknesses:**

While the proposed method is intriguing, the experimental evaluation lacks comprehensiveness. Several key recent baselines are missing, which undermines the validity of the results. The authors aim to reduce the complexity of self-attention-based graph transformers, yet they only test on a single large dataset, **ogbn-arxiv**. On this dataset, the proposed model does not outperform the vanilla **GraphSAGE** model, which is not even included in the baseline comparisons.

In **Table 1**, crucial baselines such as **Graph-ViT/MLP-Mixer** [1] and **GRIT** [2] are absent, both of which achieve significantly higher performance than the highlighted metrics. Additionally, **GECO** [3], a graph learning model based on SSMs, outperforms the presented method across datasets in **Tables 1-3**, including **ogbn-arxiv**, with significant margins.

The results in **Table 3** are not compelling due to the absence of proper GNN and GT baselines. The proposed model is designed to reduce the quadratic complexity of Graph Transformers with respect to the number of nodes \( N \) in the graph. However, aside from **ogbn-arxiv**, all other datasets have very small \( N \) values, as **Tables 1 and 2** focus on graph-level tasks. Notably, the performance on **ogbn-arxiv** is unimpressive, failing to surpass vanilla **GCN** or **GraphSAGE** models, which themselves are outperformed by **Exphormer**. Yet, **GraphSAGE** and **GCN** are missing from the evaluation.

Since this paper’s motivation is to reduce the complexity of Graph Transformers, evaluating on large-node prediction datasets is crucial to demonstrate the effectiveness of the proposed method. However, the current experiments lack both comprehensiveness and compelling experimental evidence.

Even on the smaller datasets in **Tables 1 and 2**, the results are underwhelming, with many recent baselines absent. These missing baselines include **SGFormer** [4], **Polynormer** [5], and possibly others, suggesting a need for a more extensive literature review.

- [1] **Graph-ViT/MLP-Mixer**, ICML 2023
- [2] **GRIT: Graph Inductive Biases in Transformers without Message Passing**, ICML 2023
- [3] **GECO**, [arXiv 2024](https://arxiv.org/pdf/2406.12059)
- [4] **SGFormer**, ICLR 2024
- [5] **Polynormer**, ICLR 2024

**Questions:**

Please see weaknesses.

---

> ### Author Response · Authors · 2024-11-22
>
> We sincerely thank the reviewer for their insightful feedback and comments.
>
> **Q1: Method motivation and experiments on large graphs**
>
> **A:** Please see our general comment for additional information.
>
> **Q2: Missing baselines**
>
> **A:** Thank you for your feedback. As mentioned previously, the main motivation of our work is to improve long-range communication while maintaining a low memory footprint, enabling our method to scale to large graphs. With this in mind, we compare ReHub to existing methods that evaluate long-range communication and assess their performance on larger graphs.
>
> We acknowledge the critique about Graph-ViT/MLP-Mixer and currently make the proper adjustment to produce results for large graphs. In the revised version of the paper, we will include additional evaluation results, including those from the peptides-func and peptides-struct datasets, as presented in the Graph-ViT/MLP-Mixer paper. Furthermore, we will add performance comparisons with GraphSAGE and GCN, including peak memory consumption and average epoch time.
>
>
> **Further points**
>
> > Even on the smaller datasets in Tables 1 and 2, the results are underwhelming, with many recent baselines absent. These missing baselines include SGFormer [4], Polynormer [5], and possibly others, suggesting a need for a more extensive literature review.
>
> **A:** Thank you for your feedback. Tables 1 and 2 are presented in subsection 4.1, titled "Long-range graph benchmark." We intentionally did not include SGFormer and Polynormer in our comparisons because neither of these methods explicitly focuses on long-range communication nor are they evaluated on long-range benchmarks. Our goal in this section is to compare methods that are specifically designed to address long-range communication in graphs, and as such, we excluded methods that do not emphasize this aspect. However, we appreciate your suggestion and will consider a broader comparison in future work, especially with respect to methods that explicitly target long-range communication.
>
> > Additionally, GECO [3], a graph learning model based on SSMs, outperforms the presented method across datasets in Tables 1-3, including ogbn-arxiv, with significant margins.
>
> **A:** Thank you for your comment. In lines 113-116, we acknowledge the impressive results achieved by SSM-based models, such as GECO, across various datasets. However, the focus of our paper is on transformer-based methods, which is why we chose to compare ReHub primarily to other relevant transformer-based models. We believe this approach provides a more targeted comparison within the scope of our research, but we recognize the value of exploring SSM-based models in future work.

---

### Official Review · Reviewer_rjik · 2024-11-07

**Soundness:** 3
**Presentation:** 3
**Contribution:** 2
**Rating:** 5
**Confidence:** 3

**Summary:**

This paper presents ReHub, an improved graph transformer that leverages dynamic hub reassignment to achieve linear complexity.

**Strengths:**

- The proposed method is simple
- Experimental results demonstrate that ReHub consistently achieves higher accuracy than existing SOTA methods.

**Weaknesses:**

Major:
- This paper highlights linear complexity of the proposed algorithm, but the experimental efficiency comparison is missing.
- Convergence comparison is missing. ReHub achieves higher accuracy, but it's not clear whether ReHub needs more iterations to converge.
- The datasets used in experiments are too small. The largest graph only contains 169K nodes.

Minor:
- It would be better if the authors can visualize the hub assignment to see if the proposed method generates meaningful pattern.

**Questions:**

Please address weaknesses mentioned above.

---

> ### Author Response · Authors · 2024-11-22
>
> We sincerely thank the reviewer for their insightful feedback and comments.
>
> **Q1: Experimental efficiency comparison**
>
> > This paper highlights linear complexity of the proposed algorithm, but the experimental efficiency comparison is missing.
>
> **A:** We have provided demonstrations of ReHub’s efficiency in Table 3, where we show our method requires less memory than Exphormer. Moreover, in Figure 2 we provide the peak memory of different methods on varying numbers of graph size during inference time.
>
> Additionally, we extend Table 3 and add the average epoch time during inference. The average epoch time is the wall clock time an epoch ran during our evaluation averaged throughout 5 runs with different seeds. As can be seen ReHub achieves both faster inference and lower memory consumption compared to Exphormer.
>
> | Model                  |                    | Coauthor Physics  |              |                    | OGBN-Arxiv        |              |
> |------------------------|--------------------|-------------------|--------------|--------------------|-------------------|--------------|
> |                        | Peak Memory (GB) ↓ | Avg. Epoch Time (ms) ↓ | Accuracy ↑   | Peak Memory (GB) ↓ | Avg. Epoch Time (ms) ↓ | Accuracy ↑   |
> | GraphGPS (Transformer) | OOM                | -                 | -            | OOM                | -                 | -            |
> | Exphormer              | 1.77               | 373 ± 69                 | 97.16 ± 0.13 | 2.83               | 203 ± 7                 | 72.44 ± 0.28 |
> | ReHub (Ours)           | 1.13               | 320 ± 24              | 96.89 ± 0.19 | 2.45               | 112 ± 9             | 71.06 ± 0.40 |
>
>
> **Q2: Convergence of ReHub**
>
> > Convergence comparison is missing. ReHub achieves higher accuracy, but it's not clear whether ReHub needs more iterations to converge.
>
> **A:** ReHub uses the same number of iterations as the competitors for a fair comparison.All hyperparameters, including the number of training iterations, are detailed in Appendix A.2 and can also be found in the provided config files in the attached code. The reported results in the experiments are the same as they were presented in the original papers of the compared methods. To clarify, we summarize the number of epochs used for each method below. As can be seen we have adjusted the training iterations according to prior works.
>
> | Model        | Peptides-func | Peptides-struct | PCQM-Contact | PascalVOC-SP | Coauthor Physics | OGBN-Arxiv |
> |--------------|---------------|-----------------|--------------|--------------|------------------|------------|
> | GraphGPS     | 200           | 200             | 200          | 300          | -                | -          |
> | Exphormer    | 200           | 200             | 200          | 300          | 70               | 600        |
> | NeuralAtoms  | 200           | 200             | 200          | -            | -                | -          |
> | ReHub (Ours) | 200           | 200             | 200          | 300          | 70               | 600        |
>
>
>
> **Q3: Large graph experiments**
>
> > The datasets used in experiments are too small. The largest graph only contains 169K nodes.
>
> **A:** The largest graph used in our experiments is of 700K nodes. Please see our general comment for additional information.

---

### Author Response · Authors · 2024-11-22

We sincerely thank the reviewers for their insightful feedback and comments. We appreciate the positive remarks about our method, including its simplicity, clear presentation, and wide applicability to various graph tasks with minimal modifications. We are particularly pleased that the proposed architecture and the Spokes and Hub algorithm were recognized as well-motivated and plausible for graph learning. Additionally, we are grateful for the recognition of our method’s ability to maintain linear complexity even as the number of hubs increases significantly, which highlights the efficiency of the long-range spoke update layer.

We would also like to address a common concern raised regarding the experimental evaluation on large graphs and long-range communication. The primary motivation of our work is to improve long-range communication while maintaining a low memory footprint, enabling scalability to large graphs. However, we face a challenge due to the lack of benchmarks that both focus on long-range communication and test it on large graphs. Existing benchmarks for long-range communication are typically designed for small to medium-sized graphs, while benchmarks for large graphs often do not emphasize long-range communication, as the tasks they focus on can be solved using local neighborhoods within the graph, as evidenced by the performance of local GCN methods. To address this, we follow the experimental setup used in Exphormer, showing competitive performance on long-range tasks with modest graph sizes such as peptides-func. Then, we demonstrate our efficient computational performance on larger graphs like OGBN-arxiv and coauthor physics, where, for example, Transformer-based graph models already encounter out-of-memory (OOM) issues. Additionally, we further show the scalability of our method on even larger synthetic graphs with up to 700K nodes, highlighting its ability to handle large graphs while maintaining a low memory footprint. We believe this demonstrates the efficiency of ReHub in large-scale graph tasks. We are actively exploring ways to contribute to the development of such benchmarks, which would provide a clearer evaluation of our method's strengths.

---

### Comment · Reviewer_15hB · 2024-11-26

Thank you for your detailed response. However, I will stick to my original score.

---

### Meta-Review · Area_Chair_mrNi · 2024-12-17

**Metareview:**

The paper proposes a scalable graph transformer, ReHub, that achieves (near-)linear complexity using adaptive hub reassignment. ReHub maintains global attention without the high computational cost, by assigning modes to a limited number of virtual huns and using hub-hub similarity for reassignment.

I thank the reviewers and the authors for their discussions. While the reviewers acknowledge that the proposed method is intuitive, major concerns were raised with regards to the experimental evaluation, in particular, that only small datasets were used and that crucial baselines were missing. The authors seem to acknowledge these limitations. Therefore, I recommend rejection.

**Additional Comments On Reviewer Discussion:**

While the reviewers acknowledge that the proposed method is intuitive, major concerns were raised with regards to the experimental evaluation, in particular, that only small datasets were used and that crucial baselines were missing. The authors seem to acknowledge these limitations. Therefore, I recommend rejection.

---

### Decision · Program_Chairs · 2025-01-22

Reject